# *Saccharomyces cerevisiae* Rhodanese RDL2 Uses the Arg Residue of the Active-Site Loop for Thiosulfate Decomposition

**DOI:** 10.3390/antiox10101525

**Published:** 2021-09-26

**Authors:** Qingda Wang, Huanjie Li, Yongzhen Xia, Luying Xun, Huaiwei Liu

**Affiliations:** 1State Key Laboratory of Microbial Technology, Shandong University, Qingdao 266237, China; wangqingda@mail.sdu.edu.cn (Q.W.); xiayongzhen2002@sdu.edu.cn (Y.X.); luying_xun@vetmed.wsu.edu (L.X.); 2School of Medicine, Cheeloo College of Medicine, Shandong University, Jinan 260024, China; lihuanjie@sdu.edu.cn; 3Department of Chemistry, School of Molecular Biosciences, Washington State University, Pullman, WA 99164-4630, USA

**Keywords:** rhodanese, crystal structure, thiosulfate, polysulfides, *Saccharomyces cerevisiae*

## Abstract

Persulfide, polysulfide and thiosulfate are examples of sulfane sulfur containing chemicals that play multiple functions in biological systems. Rhodaneses are widely present in all three kingdoms of life, which catalyze sulfur transfer among these sulfane sulfur-containing chemicals. The mechanism of how rhodaneses function is not well understood. *Saccharomyces cerevisiae* rhodanese 2 (RDL2) is involved in mitochondrial biogenesis and cell cycle control. Herein, we report a 2.47 Å resolution structure of RDL2 co-crystallized with thiosulfate (PDB entry: 6K6R). The presence of an extra sulfur atom S_δ_, forming a persulfide bond with the S_γ_ atom of Cys_106_, was observed. Distinct from the persulfide groups in GlpE (PDB entry:1GMX) and rhobov (PDB entry:1BOI), the persulfide group of RDL2 is located in a peanut-like pocket of the neutral electrostatic field and is far away from positively charged amino acid residues of its active-site loop, suggesting no interaction between them. This finding suggests that the positively charged amino acid residues are not involved in the stabilization of the persulfide group. Activity assays indicate that the Arg_111_ of the active-site loop is critical for the sulfane sulfur transfer. In vitro assays indicate that Arg propels the thiosulfate decomposition. Thus, we propose that Arg can offer a hydrogen bond-rich, acidic-like microenvironment in RDL2 in which thiosulfate decomposes to release sulfane sulfur. Thr of the active-site loop of rhodaneses has the same functions as Arg. Our proposal may explain the catalyzing mechanism of rhodaneses.

## 1. Introduction

Reactive sulfur species (RSS) are a group of sulfur containing compounds commonly existing in biological systems. They are essential to life due to the roles in cell signaling, redox homeostasis, and metabolic regulation [1]. Representative RSS include hydrogen sulfide (H_2_S), hydrogen persulfide (HSSH), organic persulfides (RSSH), and polysulfides (HSS_n_H, RSS_n_H, RSS_n_R, *n* ≥ 2). Among them, H_2_S has been recognized as the third gasotransmitter after carbon monoxide (CO) and nitric oxide (NO) [2]. It exerts various effects including cytoprotection, anti-inflammation, angiogenesis, and vasodilation at low concentrations [3]. However, the chemical mechanisms that H_2_S signaling remain unclear. Per/polysulfides are oxidation products of H_2_S that contain one or more zero valent sulfur atoms (S^0^, sulfane sufur). Recent studies indicated that per/polysulfides participate in H_2_S signaling [4,5].

Different from the chemically labile H_2_S and per/polysulfides, thiosulfate is quite inert at a neutral and alkaline pH. Its spontaneous oxidation to sulfate takes months and decomposition only happens at very acidic pH [6,7]. On the other hand, thiosulfuric acid (H_2_S_2_O_3_, p*K_a_*_1_ = 1.74 and p*K_a_*_2_ = 0.6) and its monoanion (HS_2_O_3_^−^) are incredibly unstable and easily decompose to sulfane sulfur and sulfite [8]. Previous studies have indicated that the reason for this is that H_2_S_2_O_3_ and HS_2_O_3_^−^ can exist as special isomers in which the negative charged O^−^ is neutralized by H^+^ to allow the formation of the S=S bond [9]. The S=S bond is scissile and one sulfur atom is easily released as free sulfane sulfur [10]. The released sulfane sulfur is prone to forming a sulfur chain or attaching itself to a nucleophile, making the decomposition reaction irreversible [7].

Thiosulfate exits as anion (S_2_O_3_^2−^) in a physiological pH range (7.2~7.6). It is usually regarded as a stable sulfane sulfur carrier in biological systems. For instance, the sulfide: quinone oxidoreductase (SQR) and persulfide dioxygenase (PDO) mediated H_2_S oxidation pathway usually produces sulfite as the final product; however, when RSSH (the product of SQR) is more abundant than the oxidation capability of PDO, sulfite and RSSH can automatically react to produce thiosulfate [11,12,13]. The fates for produced thiosulfate include being further oxidized to sulfate by the sulfur oxidation system (SOX), exported out of the cell, and decomposed by enzymes. Similarly to per/polysulfides, thiosulfate may also be related to H_2_S signaling, evidenced by the recent finding that the colonic thiosulfate level is increased during inflammatory conditions [14].

Rhodaneses, or thiosulfate: cyanide sulfurtransferases (E.C. 2.8.1.1) was first found in certain tissues of mammals and defined by the activity of catalyzing the transfer of a sulfane sulfur atom from thiosulfate to cyanide, a reaction for cyanide detoxification (Reaction 1) [15]. Further studies indicated that these enzymes are wildly distributed in all three kingdoms of life and are involved in multiple bioprocesses. For instance, the human rhodanese TST is related to Crohn’s disease and is essential for 5S ribosomal RNA import into mitochondria [16,17]. The *Saccharomyces cerevisiae* rhodanese 1 (RDL1) is critical for thiosulfate assimilation through converting thiosulfate to glutathione persulfide (GSSH) (Reaction 2) [18]. According to Reaction 2, Rhodnaese is able to utilize stable thiosulfate to generate reactive sulfane sulfur such as GSSH. Since GSSH is involved in scavenging ROS which is important for maintaining cellular redox homeostasis. It is possible that rhodanese is also involved in the antioxidant processes of cell.

S_2_SO_3_^2−^ + CN^−^ → SCN^−^ + SO_3_^2−^(1)

S_2_SO_3_^2−^ + GSH → GSSH + SO_3_^2−^(2)

More than a dozen rhodanese structures have been determined by X-ray or NMR methods. They share a universal structural model (rhodanese domain) with an α/β topology in which α helices surround a central five-stranded β-sheet [19,20]. The active site is a loop composed of six amino acid residues starting with Cys (the active-site loop) [21,22]. Except for the conserved Cys, the other five amino acid residues are widely distinct in different rhodaneses [23]. A common feature is basic amino acid residues have a high presence frequency in the active-site loop. The rhodanese domain can be present singly, in tandem repeats or fused to other protein modules [23].

Rhodaneses catalyze sulfur transfer reactions with the ping-pong mechanism. First, the sulfane sulfur of thiosulfate or other sulfane sulfur donors is transferred to the thiol group of conserved Cys to form *ES*^0^, the enzyme-sulfur adduct. Second, *ES*^0^ transfers the sulfane sulfur to a nucleophilic acceptor, such as cyanide and GSH anions (CN^−^ and GS^−^), and the occupation of Cys thiol is released [23]. Regarding this catalysis mechanism, one question has not been clearly answered: how is sulfane sulfur cut off from thiosulfate? Compared to sulfite anion (SO_3_^2−^), thiol anion (RS^−^) is a weaker nucleophile, hence the sulfane sulfur transfer from thiosulfate to Cys thiol is not chemically preferred, evidenced by the finding that GSSH can spontaneously react with sulfite to produce thiosulfate at neutral pH and 25 °C; whereas, the reversed reaction rarely happens [11,18]. Rhodanese must have a catalytical way to break the stable S-S bond of thiolsulfate.

In this study, we determined the 3D structure of *S. cerevisiae* RDL2, a single-domain rhodanese involved in mitochondrial biogenesis [24] and cell cycle control [25]. Kinetic characterization indicated that RDL2 catalyzes both Reactions 1 and 2. Mutating the last amino acid of its active-site loop (Arg_111_) resulted in loss of the capability of forming *ES*^0^ intermediate, indicating that Arg_111_ is essential for breaking S-S bond of thiolsulfate. Further studies suggested that the loop-ending amino acid, mostly Arg or hydroxyl containing-Thr/Ser [26], commonly plays such roles in rhodaneses. Thus, we proposed a mechanism for explaining how rhodaneses subtract a sulfur from a stable substrate like thiosulfate to form the unstable *ES*^0^.

## 2. Materials and Methods

### 2.1. Strains and Materials

*Escherichia coli* BL21(DE3) strain and the pET30a plasmid (purchased from Invitrogen, Shanghai, China) were used for protein expression. Thiosulfate, potassium cyanide, and reduced glutathione (GSH) were purchased from Sigma-Aldrich. Other chemicals used for *E. coli* cultivation and protein purification were purchased from (Sangon Biotech, Shanghai, China) if not specifically mentioned.

### 2.2. Protein Expression and Purification

The gene encoding RDL2 was amplified from genomic DNA of *S. cerevisiae* BY4742 as reported previously [27]. The DNA fragment encoding DUF442 domain of CpSQR was amplified from genomic DNA of *Cupriavidus pinatubonensis* JMP134. The gene encoding TST was codon-optimized for expression in *E. coli* and synthesized by Beijing Genomics institution (BGI). These genes were ligated with pET30a plasmid using the T5 exonuclease-dependent assembly method [28]. A 4 μL aliquot of 5X TEDA reaction solution was thawed on ice, and 16 μL of DNA solution including the linear pET30a and insert was added. After mixing, the reaction was carried out at 30 °C for 40 min and terminated by placing on ice, and the reaction solution was then used for transformation. Mutants of these proteins were constructed using the revised QuikChange™ method [29].

*E. coli* BL21(DE3) strains harboring the expression plasmids were incubated in LB medium at 25 °C with shaking (225 rpm). Kanamycin (50 μg/mL) was added. When OD_600_ reached to 0.6, 0.1 mM isopropyl β-d-1-thiogalactopyranoside (IPTG) was added to induce the expression, and the temperature was decreased to 16 °C. The cultivation was further continued for 22 h, and then cells were harvested by centrifugation and resuspended in buffer I (20 mM Tris-HCl, 0.5 M NaCl, 20 mM imidazole, pH 8.0). Cell disruption was performed using a Pressure Cell Homogeniser (SPCH-18) at 4 °C. Cell lysate was centrifuged to remove the debris. Target proteins in supernatant were first purified via using nickelnitrilotriacetate (Ni-NTA) agarose, obtained proteins were then passed through the size exclusion column (Superdex 200; GE Healthcare, Shanghai, China) for further purification.

### 2.3. Crystallization, Data Collection, and Structure Determination

For crystallization screens, 10 mg/mL purified protein was incubated with thiosulfate at 1:20 molar ratio. Hampton Research kits and the hanging drop vapor diffusion method were used to get preliminary crystallization conditions. RDL2 crystal was obtained after one week incubation at 20 °C in 3.5 M ammonium citrate (pH 7.0). For data collection, crystals were flash-frozen in liquid nitrogen, with 15~20% (*v*/*v*) ethylene glycol as cryoprotectant.

The X-ray diffraction data sets were collected at 100 K on a beam line BL17U at the Shanghai Synchrotron Radiation Facility (Shanghai, China) equipped with an ADSC Q315r CCD-detector. All data were processed using HKL-2000. The structure of RDL2-SSH was resolved by molecular replacement using Phaser from the CCP4 suit of programs, with RDL1 (PDB entry: 3D1P) as the search model. Refinement was performed using the PHENIX crystallography suite and the COOT interactive model-building program. The quality of final models was checked using the PROCHECK program. The structural figures were prepared with PyMOL.

### 2.4. Protein LC-MS/MS Analysis

The purified protein (6.0 mg/mL) was mixed with 200 μM of thiosulfate in HEPES buffer (100 mM, pH 7.4). After an incubation at 25 °C for 20 min, the mixture was loaded onto PD-10 desalting column to remove unreacted thiosulfate. The obtained protein sample was reacted with iodoacetamide (IAM) then digested with trypsin by following a previously reported protocol [30]. The Prominence nano-LC system (Shimadzu, Shanghai, China) equipped with a custom-made silica column (75 μm × 15 cm) packed with 3-μm Reprosil-Pur 120 C18-AQ was used for the analysis. For the elution process, a 100 min gradient from 0% to 100% of solvent B (0.1% formic acid in 98% acetonitrile) at 300 nl/min was used; solvent A was 0.1% formic acid in 2% acetonitrile). The eluent was ionized and electrosprayed via LTQ-Orbitrap Velos Pro CID mass spectrometer (Thermo Scientific, Shanghai, China) which was run in data-dependent acquisition mode with Xcalibur 2.2.0 software (Thermo Scientific). Full-scan MS spectra (from 400 to 1800 *m/z*) were detected in the Orbitrap with a resolution of 60,000 at 400 *m*/*z*.

### 2.5. RS_2_ Analysis of Thiosulfate and Proteins

The RF-5301 PC Spectrofluoro Photometer (SHIMADZU) was used for RS_2_ analysis. Thiosulfate (12.5 mM) was diluted into 2 mL argon-deoxygenated buffers (pH 3–10) in a parafilm-sealed fluorometer cell (d = 1 cm). After an incubation at 25 °C for 30 min. RS_2_ was acquired by simultaneously scanning the excitation (*λ_ex_*) and emission (*λ_em_*) on monochromators setting the offset (Δ*λ* = *λ_em_* − *λ_ex_*) to a constant as described previously [27]. All spectra were acquired with a scan rate of 60 nm/min. The measurement interval was 1.0 nm and the slit width was 5 nm.

For protein analysis, the protein (3.0 mg/mL) was incubated with thiosulfate at 1:20 molar ratio in Tris-HCl buffer (50 mM, pH 7.4). After being incubated at 25 °C for 20 min, the mixture was loaded onto a PD-10 desalting column to remove small molecules. The obtained protein was diluted to 0.1 mg/mL~0.5 mg/mL to let the RS_2_ signal intensity fall in the detection range of RF-5301 fluorometer. As the control, a protein sample without reacting with thiosulfate was also analyzed. ΔRS_2_ at a specific wavelength was calculated by RS_2_*^w^* _reacted_ − RS_2_*^w^* _control_ (w is the wavelength). Total ΔRS_2_ was calculated by adding up all ΔRS_2_ values at the 240 nm–550 nm range.

### 2.6. Thiosulfate:Cyanide Sulfurtransferase Activity Assay

The activity on Reaction 1 was measured using a colorimetric assay method described previously [31]. The reaction mixture (250 µL) contained 1 mM~300 mM thiosulfate, 1 mM~300 mM potassium cyanide, and 0.5 µg rhodanese in 300 mM HEPES, pH 7.4. The reaction was performed at room temperature for 5 min and was quenched by adding 250 µL 15% (*w*/*v*) formaldehyde, followed by addition of 500 µL ferric nitrate solution (165 mM ferric nitrate monahydrate, 13.3% (*v*/*v*) nitric acid). The absorbance of the resulting ferric thiocyanate complex was measured at 460 nm. The concentration of thiocyanate was determined using a standard curve.

### 2.7. Thiosulfate:GSH Sulfurtransferase Activity Assay

The activity on Reaction 2 was measured by detecting H_2_S formation using the lead acetate assay as described previously [32]. Briefly, the reaction mixture (1 mL) contained 0.1 mM~20 mM thiosulfate, 2 mM~40 mM GSH, 10 μg of rhodanese, and 0.4 mM leadacetate in 300 mM HEPES buffer, pH 7.4. The reaction was performed at 37 °C for 4 min. Formation of lead sulfide was measured at 390 nm. The concentration of H_2_S formed was calculated using an extinction coefficient of 5500 M^−1^ cm^−1^ for lead sulfide.

### 2.8. HPLC Analysis of the Products Generated by Thiosulfate Decomposition

Thiosulfate (500 mM) was diluted in buffers of different pH values and incubated at 25 °C for 30 min. The decomposition products were derivatized with methyl trifluoromethanesulfonate (methyl triflate) and analyzed by reversed-phase liquid chromatography using a C18 reverse phase column (VP-ODS, 150 × 4 mm, Shimadzu) and eluted with pure methanol. The peak positions of dimethylpolysulfides from Me_2_S_2_ to Me_2_S_8_ were identified using a protocol reported previously [33].

### 2.9. Detection of Sulfane Sulfurs Using SSP4

SSP4 (Sulfane Sulfur Probe 4) is a fluorescent probe to detect sulfane sulfurs specifically. SSP4 itself is non-fluorescent, but it emits strong green fluorescence after it reacts with sulfane sulfurs. Reactions of thiosulfate with SSP4 were conducted by mixing 10 μM SSP4 with 50 mM, 100 mM, or 300 mM thiosulfate in 200 μL HEPES buffer (100 mM, pH 3.0). The mixture was incubated at room temperature for 30 min, and then the fluorescence was detected by using the Synergy H1 microplate reader. The excitation wavelength was set to 482 nm and the emission wavelength was set to 515 nm.

For analysis of thiosulfate decomposition, 1 M thiosulfate was diluted into 200 μL HEPES buffer (100 mM, pH 4.0) to a final concentration of 300 mM. 100 mM Arg, Thr, or Gly was added individually. At the same time, 10 μM of SSP4 was added to incubate with thiosulfate for 30 min, and fluorescence was detected every 30 s by using the Synergy H1 microplate reader.

### 2.10. Bioinformatics Analysis and Protein Structure Modeling

The three-dimensional structure of TST was generated by SWISS-MODEL (http://swissmodel.expasy.org/, accessed on 18 July 2018) using Bos taurus rhodanese (PDB entry: 1DP2) as the template (90.7% sequence similarity). The global QMEAN score was 0.70 for the TST model. The surface electrostatic potentials were analyzed by APBS-1.1.0, and the data and parameters were obtained with the PDB2PQR server (http://nbcr-222.ucsd.edu/pdb2pqr_2.1.1/). The rhodanese pocket shape parameters were calculated using the Caver Analyst software (v2.0), whose binaries and documentation are freely available at http://www.caver.cz/.

## 3. Results

### 3.1. Crystal Structure of RDL2

The His-tag fused RDL2 was expressed in *E. coli* BL21(DE3) and purified using affinity chromatography. Its molecular weight is about 16.7 kDa judged by SDS-PAGE analysis. The crystal structure of RDL2 co-crystallized with thiosulfate was obtained at 2.47 Å resolution by molecular replacement using the RDL1 structure (PDB code: 3D1P) as template (Table 1). The final model contains two chains in the asymmetric unit (Figure 1A). Each molecule consists of a four-stranded parallel β-sheet core framed by six α-helices, displaying as a typical α/β rhodanese domain. The intermolecular interactions between two chains involve in an interface of 432 Å^2^, representing 6% of the total RDL2 surface. Only two hydrogen bonds are involved in the intermolecular interactions, suggesting that the interactions are mostly based on *van der Waals* contacts. This analysis indicates that RDL2 does not assemble as firm dimers during crystallization process.

Compared with RDL1 and TSTD1 that consist of five β-sheets and six α-helices, RDL2 is lack of β1 (Figure 1B). Structure alignment analysis demonstrates that Gln and Ser amino acid residues comprise β1 in RDL1, whereas in RDL2, these positions are occupied by Lys and Ile amino acids, comprising an unspecific shape (Figure 1C). The catalytic Cys106 stretches out β_4_ towards surface of RDL2. Electron density and bond length analysis indicates the presence of an extra sulfur atom S_δ_, forming a persulfide bond with the S_γ_ atom of Cys_106_ (Figure 1D). The persulfide group of RDL2 is located in a peanut-like pocket of neutral electrostatic field. Adjacent to it is a cradle-like pocket with positive electrostatic field, which is attributed by Arg_111_ and Arg_55_ (Figure 1E).

The active-site loop comprised by six sequential amino acids is the structural signature of rhodanese. In RDL2, the active-site loop is comprised by CAKGVR. The NH_3_^+^ groups of Arg_111_ locate at 11.3 Å away from the S_δ_ atom, and NH_3_^+^ group of Lys_108_ locates further, suggesting that they cannot form hydrogen bond with the S_δ_ atom of Cys_106_. Other neighbour residues including Ile_129_, Pro_131_, and Tyr_130_ also locate at distances beyond a hydrogen bond (Figure 1F). This analysis indicates that the persulfide group of Cys_106_ is not stabilized by specific residue(s) in RDL2.

### 3.2. Tandem MS Analysis of Thiosulfate Reacted-RDL2

We also analyzed the thiosulfate-reacted RDL2 (without a sulfur acceptor) by using LC-MS/MS. A peptide containing Cys_106_ was detected with a molecular weight of 2320.19 Da, corresponding to the Cys_106_-SSH modification (Appendix A). This modification was not detected in DTT treated RDL2. This analysis confirmed that a sulfur atom is transferred from thiosulfate to RDL2 to form the *ES*^0^ intermediate.

### 3.3. Activity Assay of RDL2

RDL2 shows typical sulfurtransferase activity evidenced by efficiently catalyzing the Reaction 1. The *K_m_* values for thiosulfate and cyanide are 32.50 ± 3.50 mM and 5.85 ± 2.18 mM, respectively, the *V_max_* is 55.04 ± 3.27 μmol min^−1^ mg^−1^ and the *k_cat_* is 15.32 s^−1^ at 25 °C. It also catalyzes the Reaction 2 and the *K_m_* values for thiosulfate and GSH are 2.54 ± 0.27 mM and 50.01 ± 8.73 mM, respectively, the *V_max_* is 0.357 ± 0.031 μmol min^−1^ mg^−1^ and the *k_cat_* is 0.099 s^−1^ at 25 °C (Figure 2 and Table 2). RDL1 and RDL2 are both sulfotransferases in *Saccharomyces cerevisiae* with similar structures. We also analyzed RDL1 activity. For Reaction 1, The *K_m_* values for thiosulfate and cyanide were 4.0 ± 0.4 mM and 2.6 ± 0.3 mM. For Reaction 2, the Km values for thiosulfate and GSH were 3.1 ± 0.3 mM and 5.6 ± 0.8 mM. These results indicate that as RDL2, RDL1 also has the sulfane sulfur transferring activity.

### 3.4. Effect of Arg_111_ on RDL2 Activity

We suspected that Arg_111_ may participate in thiosulfate anion (S_2_O_3_^2−^) or cyanide anion (CN^−^) trapping. To test this, we mutated Arg_111_ to Ile. The R111I mutant lost about 90% activity of catalysing Reaction 1. The *K_m_* values for thiosulfate and cyanide increase to 100.66 ± 55.95 mM and 60.12 ± 33.78 mM, respectively, indicating that the affinity for both of them is severely impaired. No activity on Reaction 2 was detected (Figure 2 and Table 2).

We recently reported a resonance synchronous spectroscopy (RS_2_) method for detecting both sulfane sulfur containing compounds and *ES*^0^ intermediate of rhodanese. They show strong RS_2_ signal in Uv-Vis range due to the presence of active sulfane sulfur. Thiosulfate has no significant RS_2_ signal at pH 4~10 range [27]. Compared to reduced rhodanese (Cys-SH), *ES*^0^ intermediate (rhodanese-SSH) has significant higher RS_2_ signal. Therefore, by detecting the change of RS_2_ signal, we can infer whether the rhodanese contains Cys-SSH. ΔRS_2_ reflects the change of RS_2_ signal, which is calculated by ΔRS_2_ = RS_2 reacted-rhodanese_ − RS_2 reduced rhodanese_. Total ΔRS_2_ was calculated by adding up all ΔRS_2_ values at 240 nm~550 nm range. If total ΔRS_2_ is a positive value, the reacted rhodanese should contain Cys-SSH. If total ΔRS_2_ is a lower or zero value, the reacted rhodanese should contain less or no Cys-SSH. Herein, we used this method to detect the thiosulfate-reacted RDL2. RDL2 wild type (wt) has obviously increased RS_2_ signal after reacting with thiosulfate. The total ΔRS_2_ value, which represents the RS_2_ increase degree, is 23,648.83 RI (RS_2_ Intensity) (Figure 3 and Appendix A), indicating the formation of RDL2-SSH intermediate on Cys_106_. This observation is consistent with the result of LC-MS/MS analysis. Whereas, R111I has no increased RS_2_ signal after reacting with thiosulfate (Figure 3 and Appendix A), indicating that no sulfur atom is transferred to the Cys_106_ residue of R111I. These results indicate that Arg_111_ is involved in thiosulfate trapping and sulfur transferring from thiosulfate to Cys_106_.

### 3.5. Effect of Lys_108_ on RDL2 Activity

The RDL2 active-site loop contains another basic amino acid residue Lys_108_. Its location is not in the cradle- or peanut-like pocket, suggesting a nonessential role for RDL2 activity. To test this, we mutated it to Ala. The K108A mutant showed partially decreased activity of catalyzing Reactions 1 and 2, evidenced by the higher *K_m_* values, and lower *V_max_* and *k_cat_* values than that of RDL2 wt (Figure 2 and Table 2). The thiosulfate-reacted K108A mutant shows increased RS_2_ signal (total ΔRS_2_ = 8562.64 RI) but the increasing amplitude is obviously lower than that of RDL2 wt enzyme (total ΔRS_2_ = 23,648.83 RI) (Figure 3 and Appendix A). These results indicated that although not as efficient as RDL2 wt, K108A still can form Cys_106_-SSH intermediate after reacting with thiosulfate, and still can transfer the sulfane sulfur from Cys_106_-SSH to CN^−^ or GSH. Hence, in consistent with its peripheral location, Lys_108_ is not essential for the rhodanese activity.

### 3.6. Effects of Arg_99_ on DUF442 Activity

Recently, we identified a rhodanese-like domain (DUF442) from the sulfide: quinone oxidoreductase of *C. pinatubonensis* JMP134 which can catalyze sulfane sulfur transfer reactions and whose active-site loop is comprised by CRTGTR [11]. Homology modeling analysis indicated that like Arg_111_ in RDL2, the loop-end Arg_99_ also contributes to the positive electrostatic field in DUF442. To see whether Arg_99_ is involved in thiosulfate trapping like Arg_111_ of RDL2, we constructed a R99I mutant. DUF442-R99I lost about 95% activity when catalyzing Reaction 1 (Figure 4 and Table 3). The *K_m_* value for thiosulfate (*K_m_* Donor) increases to 227.68 ± 95.64 mM, about 150-fold higher than that of DUF442 wt (1.61 ± 0.45 mM), other characteristics including *K_m_* value for KCN (*K_m_* Acc), *V_max_*, and *k_cat_* are also impaired. For Reaction 2, no activity was detected from the R99I mutant while wt showed apparent activity. These results indicate that Arg_99_ is critical for DUF442 activity.

We also used RS_2_ method to analyze DUF442 wt and the R99I mutant. DUF4422 wt showed obviously increased RS_2_ signals after reacting with thiosulfate. The total ΔRS_2_ value is 23,588.28 RI. Whereas, R99I showed no RS_2_ increase after reaction (Figure 3 and Appendix A). These results indicated that like Arg_111_ for RDL2, Arg_99_ is also involved in the process of transferring a sulfur from thiosulfate to Cys_94_.

### 3.7. Effect of the Loop-End Thr_253_ on TST Activity

TST is a tandem-domain type rhodanese locating in mitochondrion of human cell (GenBank No. NP_001257412.1). Homology modelling analysis indicated that only the second domain contains an active-site loop CRKGVT. As the position of loop-end Arg in RDL2 or DUF442, the loop-end Thr also locates in the pocket with positive electrostatic field according to the modelled structure (Appendix A). To test whether it functions similarly as the former two amino acids we mutated it to Val. The T253V mutant totally loses the activity of catalyzing Reaction 1 while TST wt shows apparent activity (Figure 4 and Table 3). Interestingly, its activity of catalyzing Reaction 2 is not significantly changed compared with that of TST wt, probably because the activity on Reaction 2 is already very low, evidenced by the low *V_max_* (0.04~0.10 μmol min^−1^ mg^−1^) and *k_cat_* values (0.02~0.06 s^−1^). We also noticed that TST wt has much higher activity toward reaction 1 than 2, evidenced by the much higher *V_max_* (176.9 vs. 0.09) and *k_cat_* values (101.7 vs. 0.055). Nonetheless, these results indicate that Thr_253_ is important for TST activity.

The RS_2_ analysis demonstrated that thiosulfate-reacted TST has obviously increased RS_2_ signal (total ΔRS_2_ = 9960.69 RI) while T253V has slightly increased (total ΔRS_2_ = 7076.96 RI) after reacting with thiosulfate (Figure 3 and Appendix A), indicating that sulfur transferring process is impaired by the T253V mutation.

### 3.8. Analysis of Active-Site Loops and Positive Electrostatic Fielding-Pockets of Structured Rhodaneses

Seeing the critical role of loop-end amino acids for rhodaneses, we analyzed all rhodanese structures that have been deposited in the PDB database (until 10 April 2021, sulfurtransferases catalyzing methionine or cystine decomposition were not included). In total, 13 structures were downloaded and categorized into different groups based on the loop-end amino acids. We found that eight of the rhodaneses contained Arg at the end of active-site loops, three contained Thr and two contained Ser (Table 4).

We observed a common feature among 13 structured rhodaneses. Their loop-end amino acids all locate in superficial pockets with positive electrostatic fields, suggesting involvement of these pockets in thiosulfate trapping or binding. To see whether these pockets have shape similarities. We measured the radius, length, and curvature parameters of these pockets and performed statistical analysis on obtained data. However, results showed that the pockets have significant variances in these parameters, especially in length and curvature (Figure 5) suggesting no obvious shape similarities among them.

### 3.9. Effects of Arg and Thr on Acidic Decomposition of Thiosulfate

It has been reported that thiosulfate is unstable in acidic solutions due to decomposition to sulfane sulfur and sulfite [6]. We used RS_2_ to study the pH effects on thiosulfate decomposition. When 12.5 mM thiosulfate was dissolved in pH 4.0~10.0 solutions at 25 °C, no RS_2_ signal was detected. Whereas, when pH ≤ 3.0 solution was used, significant RS_2_ signal was observed (Figure 6A), indicating the generation of sulfane sulfur containing products. HPLC analysis indicated that S_8_, the aggregated form of sulfane sulfur, is generated at pH 3.0, but not in pH 10.0 condition from thiosulfate decomposition (Appendix A). We also used a fluorescence based per/polysulfides detection probe SSP4 to react with decomposed thiosulfate (at pH 3.0) and observed significant fluorescent signals (Figure 6B). Together, these results validated the decomposition of thiosulfate in acidic condition.

Given the wide presence of Arg and Thr in rhodaneses active-site loops, we wondered whether they could affect thiosulfate decomposition. We added Arg into thiosulfate solution (pH = 4) and using SSP4 probe to detect the decomposition. Results indicated that the decomposition is obviously accelerated compared to that without Arg addition. The addition of Thr led to similar effects. As the control, Gly had no such effect (Figure 6C). These results indicated that the side chain groups of Arg and Thr, NH_3_^+^ and OH, are responsible for the decomposition enhancement.

### 3.10. A Proposed Model for Rhodaneses Catalyzed Thiosulfate Decomposition

The 3D structure data indicate that Arg/Thr are spatially near to the conserved cysteine. Activity assay experiments indicate that they are critical for catalyzing the sulfane sulfur transfer. In vitro chemical reactions indicate they can propel thiosulfate decomposition. These results form an evidence chain and, based on this chain, we proposed a model for explaining how sulfane sulfur is transferred from stable thiosulfate to unstable *ES*^0^ at physiological pH (Figure 7). First, S_2_O_3_^2−^ is attracted by and trapped in the positive electrostatic-fielding pocket that locates on the surface of rhodaneses. Second, NH_3_^+^ group of Arg or OH group of Thr/Ser can donate shared hydrogen bond(s) to S_2_O_3_^2−^, which makes S_2_O_3_^2−^ exist as H_2_S_2_O_3_ or HS_2_O_3_^−^ form in the pocket, and hence leads to decomposition of the latter two. Third, the released sulfane sulfur from the decomposition reaction is attached to the Cys-SH group of the active-site to form Cys-SSH, and consequently the *ES*^0^ forms. 

## 4. Discussion

A pocket with positive electrostatic field near or surrounding the catalysis Cys is found in all reported rhodanese structures. This positive electrostatic field is attributed to basic or hydroxyl amino acids. It has been hypothesized that these amino acid residues participate in binding thiosulfate in the correct orientation to make its S-S bond close to the Cys thiol. After the sulfane sulfur transfer, these amino acid residues can stabilize the Cys-SSH and allow approach of the sulfur acceptor [21,26]. In this study, we obtained crystal structure of RDL2-SSH, the *ES*^0^ state of *S. cerevisiae* mitochondrial rhodanese. The persulfide group is not located in the pocket of the positive electrostatic field, and its S_δ_ atom is far away from the amide groups of Arg_111_ or Lys_108_. The location does not support the hypothesis that Arg_111_ can stabilize Cys_106_-SSH. We also performed a survey on structured rhodaneses and found that the amino acid composition of their active-site loops and the shapes of positive-electrostatic-fielding pockets have unneglectable variances. Therefore, the proposal that basic amino acids can orientate thiosulfate to a specific position in the pocket is also questionable.

Chemical analysis indicated that thiosulfate is prone to decompose in pH ≤ 3 conditions due to the formation of H_2_S_2_O_3_ or HS_2_O_3_^−^. Its decomposition leads to the production of sulfane sulfurs and sulfite. We observed that Arg and Thr can accelerate the decomposition reaction, most probably because they can offer shared H-hydrogen bonds to neutralize O^−^ anions, propelling the formation of H_2_S_2_O_3_ or HS_2_O_3_^−^. Combined with the finding that Arg and Thr/Ser are prevalent loop-ending amino acids in rhodaneses, we propose that Arg or Thr/Ser in the catalysis pocket does not orientate thiosulfate to near the Cys thiol but offer a microenvironment to allow the formation of H_2_S_2_O_3_ or HS_2_O_3_^−^, which spontaneously decomposes. It is noteworthy that the catalysis Cys thiol may also contribute to the microenvironment. We mutated the Cys_106_ of RDL2 to Ser_106_ and found that this mutant lost the catalysis activity on both Reactions 1 and 2. The Cys mutation in DUF442 led to the same result (data not shown), suggesting Cys thiols do not only function as sulfur acceptors in rhodaneses.

Rhodaneses’ substrates are not limited to thiosulfate. In a very recent work, we found that the rhodaneses-like domain DUF442 can also react with GSSH to form the *ES*^0^ intermediate at pH 7.4 [27]. The p*K*_a_ value of GSSH was determined to be 6.9; therefore, at pH 7.4 it exists mainly as relatively stable deprotonated form (GSS^−^). At protonated form GSSH is prone to release its sulfane sulfur [27]. The rapid reaction of GSSH with DUF442 at pH 7.4 suggests that the DUF442 catalytic pocket propels the protonation of GSSH. Therefore, rhodaneses impel the sulfane sulfur release from a donor substrate via providing a hydrogen bond-sharing microenvironment.

It is noteworthy that the rhodaneses tested herein all show much higher catalytic rates on Reaction 1 than on Reaction 2. This should be caused by the structural difference of acceptors. Cyanide is a much smaller compound than GSH, even smaller than thiosulfate, hence it should be able to enter the pocket and is more accessible to the persulfide group of *ES*^0^. In addition, other compounds present in biological systems, such as cysteine, homocysteine, and nitric oxide, also can be acceptors. After obtaining sulfane sulfur, they become molecules with important physiological functions. The promiscuity on both donor and acceptor substrates suggests that rhodanese play central roles in sulfane sulfur distribution, and hence are related to various functions of RSS.

## 5. Conclusions

In this study, we reported the 3D structure of rhodanese Rdl2 with a 2.47 Å resolution. From the location of its sulfane sulfur atom and activity assay results, we found that Rdl2 catalyzes thiosulfate decomposition via providing a hydrogen bond-sharing microenvironment, in which S_2_O_3_^2−^ becomes protonated-like. The Arg residue in its active-site loop is essential for Rdl2 activity. We also analyzed other rhodaneses including DUF442 and TST and finally proposed a model to explain how they cut off the sulfane sulfur cut off from thiosulfate. Our model completes the catalyzing mechanism of rhodaneses.

## Figures and Tables

**Figure 1 antioxidants-10-01525-f001:**
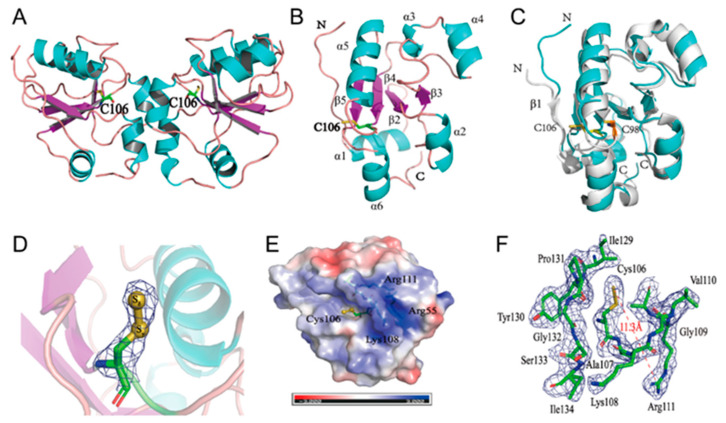
Crystal structure of RDL2 (*S. cerevisiae* rhodanese 2). (**A**) The X-ray crystal structure of RDL2 was solved at 2.47 Å by molecular replacement using the RDL1 structure (PDB entry: 3D1P) as a template. The structure of RDL2 has two asymmetric chains. (**B**) The monomer of RDL2 consists of a four-stranded parallel β-sheet core (purple) surrounded by six α-helices (blue). Cys_106_ is shown in stick representation. (**C**) Alignment of RDL2 structure (blue) with RDL1 (*S. cerevisiae* rhodanese 1) structure (white). (**D**) The electron density map of active site cysteine. The persulfide bond is formed by the S_γ_ atom of Cys_106_ with a S_δ_ atom. The electron density map (2Fo-Fc) is contoured at 1.5 σ. (**E**) The surface electrostatic potential representation of RDL2. Cys_106_ is located in a peanut-like pocket of neutral electrostatic field. Arg_111_, Arg_55_ and Lys_108_ in the vicinity of the persulfide group form a positive electrostatic field. Positive and negative electrostatic potentials are shown in blue and red, respectively in the range of ±3 kT/e. (**F**) The distance between NH_3_^+^ group of Arg_111_ and S_δ_ atom of Cys_106_ is shown in red line. The distances between S_δ_ atom and neighboring residues all exceed the hydrogen bond range.

**Figure 2 antioxidants-10-01525-f002:**
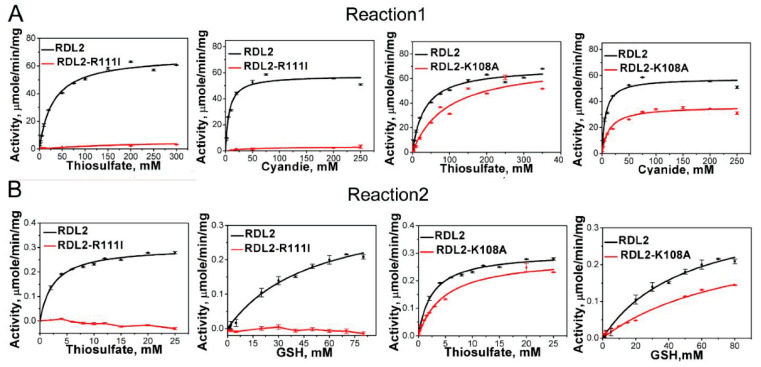
Kinetics of RDL2-catalyzed Reactions 1 (**A**) and 2 (**B**). The data shown as average ± s.d. are all from 3 independent experiments and fitted with Michaelis-Menten equation.

**Figure 3 antioxidants-10-01525-f003:**
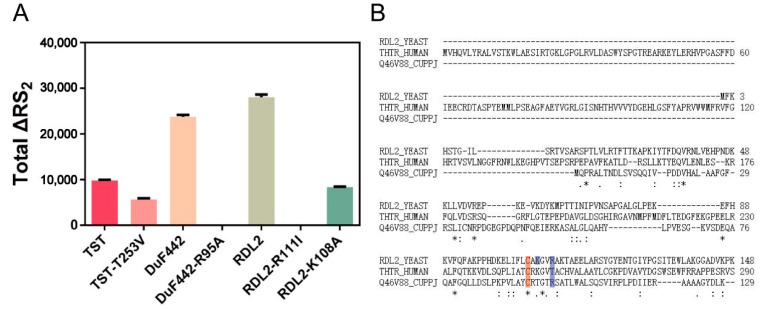
Total ΔRS_2_ values of enzymes and their mutants. (**A**) After reacting with thiosulfate, TST, DUF442, and RDL2 show positive total ΔRS_2_ values, indicating they form Cys-SSH intermediates; TST-T253V and RDL2-K108A mutants show decreased total ΔRS_2_ values compared to wt, indicating they contain less Cys-SSH than corresponding wt; DUF442-R95A and RDL2-R111I show zero ΔRS_2_ values, indicating they contain no Cys-SSH. (**B**) Sequence alignment of the rhodaneses. The conserved cysteine residue and mutation sites are shaded in color. * indicates the conserved residue.

**Figure 4 antioxidants-10-01525-f004:**
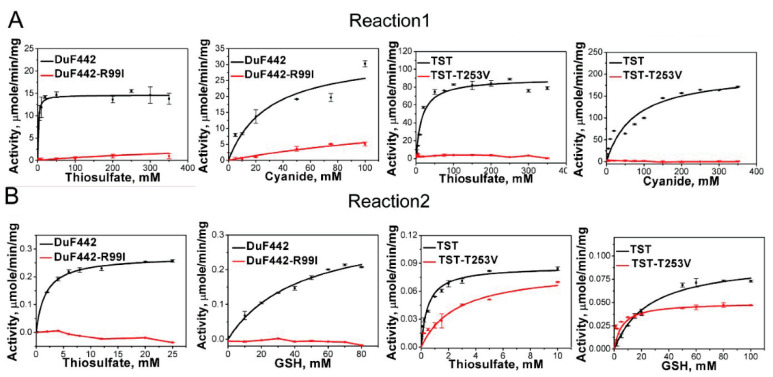
Kinetics of DUF442 and TST-catalyzed Reactions 1 (**A**) and 2 (**B**). The data shown as average ± s.d. are all from 3 independent experiments and fitted with Michaelis-Menten equation.

**Figure 5 antioxidants-10-01525-f005:**
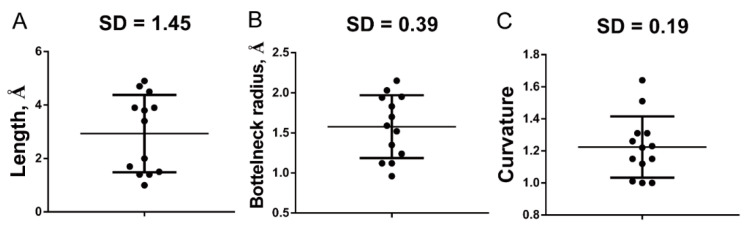
Shape parameters of the catalyzing pockets of structured rhodaneses. The rhodanese structures were downloaded from PDB database (Appendix A). The length (**A**), bottleneck radius (**B**) and curvature (**C**) parameters of their catalyzing pockets were calculated using the Caver Analysis software (v2).

**Figure 6 antioxidants-10-01525-f006:**
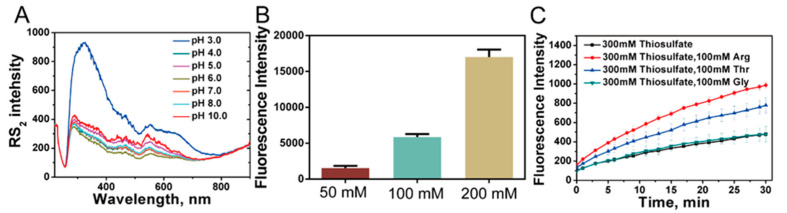
The effects of Arg and Thr on spontaneous decomposition of thiosulfate. (**A**) Thiosulfate was dissolved in pH buffers of different pH (3.0–10.0). RS_2_ was measured after a 30 min incubation at 25 °C. (**B**) Thiosulfate was reacted with SSP4 at pH 3.0 condition. (**C**) Thiosulfate was reacted with SSP4 at pH 4.0 condition, with or without presence of Arg, Thr or Gly.

**Figure 7 antioxidants-10-01525-f007:**
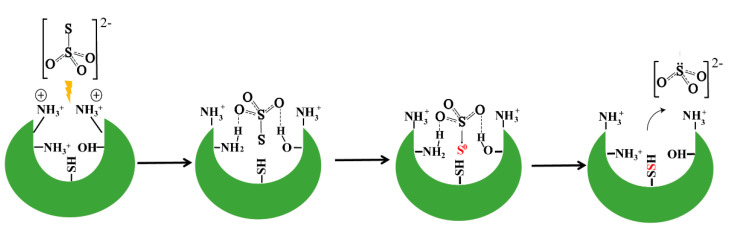
A proposed mechanism for sulfane sulfur transfer from thiosulfate to rhodanese.

**Table 1 antioxidants-10-01525-t001:** X-ray data collection and refinement statistics.

Data Collection	
Space group	*I 4* _1_
Cell dimensions	
*a*, *b*, *c* (Å)	79.753, 79.753, 110.382
α, β, γ (°)	90.00, 90.00, 90.00
Wavelength (Å)	0.9791
Resolution (Å)	50.00–2.47 (2.51–2.47)
〈*I*/σ (*I*)〉	24.69 (3)
Completeness (%)	100.0 (99.7)
Redundancy	13.1 (10.3)
CC_1/2_	0.990 (0.976)
*R_pim_*	0.028 (0.230)
**Refinement**	
Resolution (Å)	27.595–2.471
No. reflections	11,448
*R*_work_/*R*_free_ (%)	20.77/24.72
No. atoms	
Protein	1952
Water	57
*B*-factors	
Protein	35.37
Water	32.55
RMS deviations	
Bond lengths (Å)	0.002
Bond angles (°)	0.516
Ramachandran Plot	
favored	99.15%
allowed	0.85%
outliers	0.00%

**Table 2 antioxidants-10-01525-t002:** Kinetic parameters of RDL2-catalyzed Reactions 1 and 2.

	Rhodanese	*K_m_* Donor mM	*K_m_* Acc. mM	*V_max_* μmol min^−1^ mg^−1^	*k_cat_* s^−1^	*k_cat_*/*K_m_* Donor M^−1^ s^−1^	*K_cat_*/*K_m_* Acc. M^−1^ s^−1^
Reaction 1	RDL2	32.5 ± 3.50	5.85 ± 2.18	55.04 ± 3.27	15.32	0.47 × 10^3^	2.62 × 10^3^
	RDL2-R111I	100.66 ± 55.95	60.12 ± 33.78	3.40 ± 0.61	0.95	9.43	15.8
	RDL2-K108A	93.23 ± 2.67	9.13 ± 3.78	31.31 ± 2.21	8.71	93.42	0.95 × 10^3^
Reaction 2	RDL2	2.54 ± 0.27	50.01 ± 8.73	0.357 ± 0.031	0.099	38.98	1.98
	RDL2-R111I	--	--	--	--	--	--
	RDL2-K108A	4.79 ± 0.57	69.41 ± 13.51	0.153 ± 0.016	0.042	8.77	0.61

Note: For reaction 1, *K*_*m*_ Donor is assayed using fixed KCN concentration and different thiosulfate concentrations. *K*_*m*_ Acc is assayed using fixed thiosulfate concentration and different KCN concentrations. For reaction 2, *K*_*m*_ Donor is assayed using fixed GSH concentration and different thiosulfate concentrations. *K*_*m*_ Acc is assayed using fixed thiosulfate concentration and different GSH concentrations. Data shown as average ± s.d. are all from 3 independent experiments. The mutations that losing activity (--) is confirmed with two independent experiments.

**Table 3 antioxidants-10-01525-t003:** Kinetic parameters of DUF442 and TST.

	Rhodanese	*K*_*m*_ Donor mM	*K*_*m*_ Acc. mM	*V*_*max*_ μmol min−1 mg−1	*k*_*cat*_ s−1	*k*_*cat*_/*K*_*m*_ Donor M−1 s−1	*K*_*cat*_/*K*_*m*_ Acc. M−1 s−1
Reaction 1	DUF442	1.61 ± 0.45	13.27 ± 0.91	21.18 ± 0.86	4.89	3.04 × 103	0.37 × 103
	DUF442-R99I	227.68 ± 95.64	132.2 ± 28.42	8.12 ± 0.76	1.88	6.77	14.22
	TST	15.68 ± 2.86	26.59 ± 2.34	176.90 ± 3.39	101.73	6.49 × 103	3.83 × 103
	TST-T253V	--	--	--	--	--	--
Reaction 2	DUF442	1.53 ± 0.35	44.7 ± 7.41	0.334 ± 0.026	0.078	50.98	1.74
	DUF442-R99I	--	--	--	--	--	--
	TST	0.49 ± 0.10	25.62 ± 3.94	0.096 ± 0.005	0.055	0.11 × 103	2.15
	TST-TV	2.43 ± 0.70	1.77 ± 0.53	0.044 ± 0.002	0.025	10.29	14.12

Note: For reaction 1, *K*_*m*_ Donor is assayed using fixed KCN concentration and different thiosulfate concentrations. *K*_*m*_ Acc is assayed using fixed thiosulfate concentration and different KCN concentrations. For reaction 2, *K*_*m*_ Donor is assayed using fixed GSH concentration and different thiosulfate concentrations. *K*_*m*_ Acc is assayed using fixed thiosulfate concentration and different GSH concentrations. Data shown as average ± s.d. are all from 3 independent experiments. The mutations that losing activity (--) is confirmed with two independent experiments.

**Table 4 antioxidants-10-01525-t004:** Structured rhodaneses downloaded from the PDB database.

Cluster	Name	Organism	PDB ID	Loop Sequence
Arg	TSTD1	*Homo sapiens*	6BEV	CQMGKR
	RDL1	*Saccharomyces cerevisiae*	3D1p	CASGKR
	YgaP	*Escherichia coli*	5HPA	CQAGKR
	YnjE	*Escherichia coli*	2WLR	CGTGWR
	SACOL1807	*Staphylococcus aureus*	3IWH	CAGGVR
	Bphyt_4191	*Paraburkholderia phytofirmans*	5VE3	CRAGGR
	Rv0390	*Mycobacterium tuberculosis*	2FSX	CRSGNR
	TVG0868615	*Thermoplasma volcanium*	3GK5	CAHGNR
Thr	MST	*Homo sapiens*	4JGT	CGSGVT
	MST	*Leishmania major*	1OKG	CGSGVT
	TST	*Bos taurus*	1BOH	CRKGVT
Ser	GlpE	*Escherichia coli*	1GMX	CYHGNS
	TUM1	*Saccharomyces cerevisiae*	3UTN	CGTGVS

## Data Availability

The 3D structure of Rdl2 has been deposited in PDB with entry 6K6R. Other data is contained within the article or supplementary material.

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
