# Peer review of "Saccharomyces cerevisiae Rhodanese RDL2 Uses the Arg Residue of the Active-Site Loop for Thiosulfate Decomposition"

_antioxidants, 2021, doi:10.3390/antiox10101525_

Round 1

Reviewer 1 Report

The paper, written by Wang and co-workers, is a nice biochemical paper and describes the importance of an arginine residue for the catalytic function of RDL2 in yeast. The research question and the methodological approach are very clear, the data are well presented and discussed. The only thing I would like to see is that the abbreviations used in Figure 1 are explained again in the figure text for better comprehension (so that the figure is self-explanatory) and in the material and methods section it would be helpful to have a little more detail of the cloning strategy, although this has been referenced. In addition, a brief statement of what SSP4 is for would be helpful in the material and method

Author Response

We appreciate the positive feedback from the reviewer. Taken these suggestions, we have added corresponding descriptions into the figure text, materials and methods. The changes have been highlighted in yellow in the revised manuscript. 

Reviewer 2 Report

Wang et al present a work entitled “Saccharomyces cerevisiae Rhodanese RDL2 uses the Arg residues of the active-site loop for thiosulfate decomposition”. The authors combined structural analysis and activity measurements to study and compare the role of specific residue of the active site in the thiosulfate sulfurtransferase activity of some rhodanese isoforms (yeast and human, more particularly). The authors notably propose a role of arginine/threonine residue in thiosulfate decomposition before the subsequent persulfidation of conserved cysteine residue. The manuscript has major shortcomings both in terms of organization and in the conclusions drawn from the results obtained. The results are too preliminary and important controls are lacking.

Major comments

  1. The structure of this article is difficult to understand and to follow. Starting the article by describing 3D structure data might be more logical to introduce all other results and why the authors focused their attention on other proteins and on specific residues.

  1. The activity measurements are grouped in figure 1 and table 1. They include results for proteins that are only described at the end of the manuscript. The table is really hard to read. Considering these data, the obtained values are not compared to those obtained with other rhodanese isoforms or to other results obtained by other groups for human TST for example. It seems important considering that the data described here are not in the same range as those obtained before (Libiad et al, 2015, ref 20). A comparison between yeast RDL1 and RDL2 may be also useful. From a more general point of view, the results are not enough discussed. What can be the physiological relevance of such high Km values?

I also have a last comment about activity measurements. In the legend of figure 1, it is indicated “the date are from 2-3 independent experiments…” Two independent experiments are not sufficient.

  1. A second major issue concerns the comparison between yeast RDL2, human TST and the domain DUF442. First of all, an alignment of these protein sequences might facilitate the identification of the position of residues that have been mutated. From my point of view, despite DUF442 seems to display TST activity, it is not a rhodanese protein. It does not possess the conserved DxR motif at Nter part of the protein. I do not think that it is relevant to compare DUF442 and RDL2 and the putative role of arginine residue. I do not think that comparing a protein with one rhodanese domain (RDL2) and another protein with two rhodanese domain (human TST) is helpful to identify the importance of active-site residues as the protein structure is different as one protein possesses an additional domain that is important for its activity.

  1. Section 3.2 The authors have to show the data obtained for reduced RDL2 but also for thiosulfate-treated RDL2 that is subsequently reduced by DTT to confirm the reversibility of the modification and hence the persulfidation of Cys. Similar experiments would be interesting to do with DUF442 and human TST.

  1. Understanding the “RS2” experiments and associated results is not so easy. Figure 3 shows differences of RS2 between untreated and thiosulfate-treated proteins. How did the authors calculate these differences? Did they substract specific signals and did they consider all the spectra shown as supplemental figures? It is important do describe more carefully this point. For most of proteins, considering the spectra, the differences are really low. I’m not sure that they can be significant. Moreover, considering the variant RDL2 R111l, this protein displays a significant higher signal in the absence of thiosulfate. What does it mean? Finally, a control with proteins without catalytic cysteine is mandatory for each candidate.

  1. The legend of figure 3 does not correspond to the figure. Table 3 (line344) is cited but there is no table 3 in the manuscript. Please, check more carefully the manuscript before submission.

  1. Line 342. “13 structures were downloaded…” Accession number corresponding to these structures are not cited. For my opinion, this information has to be provided to help the readers to understand the analysis and to enable the readers to do themself their own point of view..

  1. The study of thiosulfate decomposition is interesting but the experiments were done using pH conditions and amino acid concentrations that are not physiologically relevant. Furthermore, it can be expected that the chemical behaviours of isolated amino acid and similar residues located in proteins are really different. From my pint of view, the obtained results do not enable to conclude about catalytic mechanism of thiosulfate reduction by proteins.

Reviewer 3 Report

The paper by Liu and colleagues reports a study on the structure and the mechanism of action of rhodaneses, which represent widespread enzymes catalysing the transfer of a sulfane sulfur atom from thiosulfate to cyanide. The S. cereviasiae rhodanese RDL1 is also involved in the sulfate assimilation by converting thiosulfates into glutathione persulfide. In this work, the authors focused their attention on the determination of the 3D structure of S. cereviasiae RDL2 as well as on the structure-related mechanism possibly accounting for the removal of sulfane sulfur from stable thiosulfates. The article claims to propose a new mechanism where Arg111 plays a central role by offering ah hydrogen bond-rich, acidic microenvironment assisting the decomposition of thiosulfates. This hypothesis is supported by in vitro results suggesting that Arg promotes the releasing of sulfane sulfur from thiosulfate under acidic conditions. Furthermore, mutating Arg111 of the active-site loop of RDL2 led to the loss of the capability to form the ES0 intermediate, thus indicating the central role of Arg111 in breaking the S-S bond of thiosulfates. The Thr residue of the active-site loop plays a similar role as Arg. The topic is interesting and the results presented herein might be useful to further elucidate the mechanism of action of rhodaneses. However, there are some issues that should be considered in order to render this work suitable for publication.

  • As aforementioned, Arg and Thr/Ser in the catalytic pocket have been proposed to provide a microenvironment allowing the decomposition of thiosulfate. However, apart from results presented in the section 3.9, the authors did not provide evidences about this statement. This point should be considered and the hypothesis (without additional experiments, this microenvironment-related effect it is an hypothesis) should be better discussed. A mechanistic explanation of the effect of Arg, Thr/Ser and Cys should be proposed. A Scheme related to this chemistry would help to illustrate the proposed mechanism. Results related to Cys mutation in DUF442 should also be presented and commented. The authors report that Cys mutation in DUF442 leads to loss of the catalytic activity. Do this mean that decomposition of thiosulfate is not observed? This seems to be in contrast with results presented in section 3.9.
  • The mechanism reported in the Figure 6 is not completely clear: i) There is a positive charge that is not localised; ii) The interactions of thiosulfate in the active-site are not clear; iii) All products formed by the sulfane sulfur transfer should be indicated in the mechanism (see the last step). I recommend to redraw the Figure 6 indicating the molecular structure of thiosulfate and clearly highlighting its interactions with residues of the active-site. All products (not only Cys-SSH formation) should be indicated.
  • Line 385: The 3 of ammonium ion should be subscripted.
  • Rhodaneses should be better presented in the field of antioxidants and their antioxidant-related aspects should be highlited in the introduction.

  • There are typing and grammar errors that should be corrected throughout the whole manuscript. Some sentences need to be rewritten as they are unclear or grammatically incorrect. Please, carefully check throughout the manuscript. Please, also ensure that all acronyms have been properly defined.

Round 2

Reviewer 2 Report

I’m sorry but I disagree with the authors about considering DUF442 as a rhodanese. As mentioned in my first review and in this paper (PMID: 31055601), it does not possess all elements characteristic of a rhodanese domain. It there was the case, it would be not named DUF. Studying this protein due its ability of using thiosulfate as substrate is of interest but if cannot be considered as true rhodanese protein. Please modify the text accordingly.

Author Response

Taken this suggestion, we changed to “Recently, we identified a rhodanese-like domain (DUF442) from the sulfide:quinone oxidoreductase of C. pinatubonensis JMP134, which can catalyze sulfane sulfur transfer reactions and whose active-site loop is comprised by CRTGTR” to describe DUF442 in the revised manuscript.